# Development of a Multienzyme Isothermal Rapid-Amplification Lateral Flow Assay for On-Site Identification of the Japanese Eel (*Anguilla japonica*)

**DOI:** 10.3390/foods14173100

**Published:** 2025-09-04

**Authors:** Eun Soo Noh, Chun-Mae Dong, Hyo Sun Jung, Jungwook Park, Injun Hwang, Jung-Ha Kang

**Affiliations:** Biotechnology Research Division, National Institute of Fisheries Science, 216, Gijanghaean-ro, Gijang-eup, Gijang-gun, Busan 46083, Republic of Korea; ehdcnsao@naver.com (C.-M.D.); jhs3010@korea.kr (H.S.J.); jjuwoogi@korea.kr (J.P.); astraroth@korea.kr (I.H.); genetics@korea.kr (J.-H.K.)

**Keywords:** *Anguilla japonica*, food authenticity, multienzyme isothermal rapid amplification, species identification

## Abstract

Eel populations are globally threatened by overfishing and illegal trade, making accurate species identification essential for resource conservation and regulatory enforcement. Conventional molecular identification methods are generally applied in the laboratory, with limited rapid on-site application. This study developed a field-deployable assay to identify the Japanese eel (*Anguilla japonica*), by incorporating multienzyme isothermal rapid amplification (MIRA) technology with a visually readable lateral flow assay (LFA). Species-specific primers targeting a 286 bp region within the mitochondrial genome of A. japonica were designed and labeled with fluorescein amidite and biotin, respectively. The performance of the MIRA-LFA was validated by assessing its specificity against four other major eel species and its analytical sensitivity, i.e., limit of detection (LoD), under optimized temperature and reaction-time conditions. The MIRA-LFA demonstrated 100% specificity, generating a positive signal only for *A. japonica*, with no cross-reactivity. A clear visual result was obtained within 10 min at the optimal reaction temperature of 39 °C. Under these optimal conditions, the assay showed a high sensitivity, with an LoD of 0.1 ng/μL of genomic DNA. The proposed assay is an effective tool for the rapid, specific, and sensitive identification of *A. japonica*. The ability to obtain fast, equipment-free visual results makes this assay an ideal point-of-care testing solution to combat seafood fraud and support the sustainable management of this economically important and vulnerable species.

## 1. Introduction

Freshwater eels (*Anguilla* spp.) are a significant fishery resource with high economic and cultural value, particularly in Asia [1]. However, global eel populations are facing a severe crisis due to a combination of factors, including overfishing, habitat destruction, and climate change [2,3]. In response to this resource depletion, the international community has strengthened conservation efforts [4]. For example, the International Union for Conservation of Nature (IUCN) has classified the European eel (*A. anguilla*) as critically endangered, and the American eel (*A. rostrata*) and Japanese eel (*A. japonica*) as endangered. Notably, the European eel is listed in Appendix II of the Convention on International Trade in Endangered Species of Wild Fauna and Flora (CITES), which strictly controls its international trade [5,6,7].

Despite these regulatory efforts, illegal, unreported, and unregulated (IUU) fishing and international smuggling of eels persist [8]. Artificial seed production technology has not yet been commercialized, such that the aquaculture industry is entirely dependent on wild-caught juveniles (glass eels). Consequently, regulated species are often fraudulently sold as unregulated species, and illegally caught glass eels frequently enter aquaculture farms. Numerous molecular studies have consistently reported that CITES-regulated European eels are widely distributed in Asian markets, disguised as unregulated Japanese eels [9,10]. This serious problem undermines conservation efforts, disrupts the fair market, and erodes consumer confidence.

In the eel production industry, effective regulatory enforcement and resource management require accurate species identification. However, morphological identification methods are unreliable, particularly at the glass eel stage, when interspecific features are indistinct [11,12]. To overcome this limitation, DNA-based molecular techniques have been developed [13,14,15]. Sequence analysis techniques such as DNA barcoding are considered the gold standard in terms of accuracy, but are unsuitable for field application due to the need for long analysis times, expensive equipment, and skilled personnel [16]. While various polymerase chain reaction (PCR)-based techniques such as species-specific, multiplex, and real-time PCR are faster than sequencing, they still require expensive thermal cyclers and post-processing steps such as electrophoresis, making them difficult to use outside the laboratory [17]. The use of real-time PCR equipment onboard ships has been attempted, but the reliance of this technique on sophisticated instrumentation remains limiting [18]. The field application gap created by these technological limitations could be filled by the development of rapid, simple, on-site diagnostic tools for use by regulators at customs offices, seafood markets, and aquaculture farms [17]. Therefore, isothermal nucleic acid amplification technologies have emerged as a groundbreaking alternative for on-site diagnostics [19,20]. Among these, recombinase polymerase amplification (RPA) rapidly amplifies DNA at low, constant temperatures of 37–42 °C using a recombinase enzyme [21].

In this study, we developed and validated a novel diagnostic system for the rapid, specific, and sensitive detection of *A. japonica* DNA by combining two existing technologies. First, we employed multienzyme isothermal rapid amplification (MIRA), an isothermal amplification technology based on a principle similar to that underlying RPA, but using proprietary enzymes and probe systems for enhanced specificity and stability [22]. These features are particularly advantageous for on-site applications, offering potentially greater reagent robustness against fluctuating field conditions and a reduced risk of non-specific amplification, thereby increasing the reliability of point-of-care diagnostics [23]. Then, to visualize the amplified products, we coupled the MIRA system with a lateral flow assay (LFA), a simple, inexpensive, paper-based test that functions in a similar manner to a human pregnancy test [24]. LFA allows for clear, equipment-free visual interpretation of assay results, making it ideal for point-of-use testing.

The combination of RPA and LFA has already demonstrated its utility in various food authentication applications [25,26]. Therefore, the primary objective of this study was to develop the proposed MIRA-LFA as a reliable tool that bridges the gap between laboratory analysis and on-site enforcement. Its significance lies in empowering regulators, distributors, and aquaculturists that will contribute to the sustainable use of eel resources and the establishment of a fair market.

## 2. Materials and Methods

### 2.1. Design of Species-Specific Primers

To develop an MIRA assay specific to *A. japonica*, complete mitochondrial (mtDNA) genome sequences of all known species within the *Anguilla* genus were retrieved from the National Center for Biotechnology Information (NCBI) GenBank database. The collected sequences were subjected to multiple sequence alignment using BioEdit v7.2 software to identify a sequence region that is conserved within *A. japonica* but is distinct from those of other *Anguilla* species (Appendix A).

Based on the alignment results, an MIRA primer pair was designed using the complete *A. japonica* mtDNA genome sequence (accession no. AB038556.2) as a reference. Specifically, the forward primer was designed to target the region from 2917 to 2951 base pairs (bp), and the reverse primer was designed to target that from 3173 to 3202 bp, yielding an expected amplicon size of 286 bp. These primers were designated AJ-MIRAF and AJ-MIRAR, respectively.

To minimize cross-reactivity, the 3′ end of the primers, which is crucial for amplification efficiency, was mismatched with sequences from non-target species. The theoretical specificity of the designed primers was verified using the NCBI Primer-BLAST program. For LFA detection, the 5′ ends of the forward and reverse primers were labeled with fluorescein amidite (FAM) and biotin, respectively (Table 1).

### 2.2. Eel Samples and Genomic DNA (gDNA) Extraction

To validate the specificity of the assay, tissue samples of five standard eel species (*A. japonica*, *A. anguilla*, *A. rostrata*, *Anguilla bicolor pacifica*, and *Anguilla marmorata*) were obtained from the National Institute of Fisheries Science (Busan, Republic of Korea) for use as positive and negative controls (Table 2). We extracted gDNA from approximately 20–25 mg of muscle tissue from each sample using a DNeasy Blood and Tissue Kit (Qiagen, Hilden, Germany), according to the manufacturer’s instructions. The concentration and purity of the extracted gDNA were determined using a NanoDrop Spectrophotometer (Thermo Fisher Scientific, Waltham, MA, USA) by measuring the 260/280 absorbance ratio. For consistency, all DNA samples were normalized to a final concentration of 10 ng/μL and stored at −20 °C.

### 2.3. MIRA-LFA Procedure

MIRA isothermal amplification was performed using a commercial kit (Amp-Future Biotech, Changzhou, China). The 25 μL reaction mixture consisted of 15 μL of buffer A, 1.5 μL of AJ-MIRAF/AJ-MIRAR primer mix (10 μM each), 1 μL of template gDNA (10 ng/μL), and 6.5 μL of nuclease-free water. All components were mixed in a tube, and then 1 μL of buffer B was added to initiate the reaction, followed by incubation at the optimal temperature for the optimal reaction time.

Upon completion of the amplification reaction, 2 μL of the MIRA product was applied to the sample pad of a commercial LFA strip (Amp-Future Biotech, Changzhou, China). According to the manufacturer, this strip is specifically designed to detect amplicons dually labeled with FITC/6-FAM and Biotin. This was followed by the addition of 78 μL of running buffer. As the liquid migrated along the strip, the FAM-labeled end of the amplicon would bind to anti-FAM antibody-conjugated gold nanoparticles. This entire complex was then captured by streptavidin immobilized on the test (T) line via the biotin-labeled end. The strip was allowed to react at ambient temperature for 2 min, and the results were read visually. The outcome was determined by the appearance of control (C) and test (T) lines on the LFA strip. The C line is an internal control that confirms proper functioning of the strip and must appear in all valid tests, and the T line indicates the presence of the *A. japonica*-specific amplicon, which is dually labeled with FAM and biotin. Thus, a result was considered positive if both the C and T lines appeared, and negative if only the C line appeared. Any test in which the C line did not appear was considered invalid and repeated.

### 2.4. Assay Performance and Validation

The performance of the MIRA-LFA was evaluated by sequentially assessing its specificity, optimal reaction conditions, and sensitivity. To confirm its specificity, the assay was tested for cross-reactivity using 10 ng of gDNA from each of the five standard eel species, including *A. japonica*. To establish the optimal reaction conditions for field application, the reaction temperature was optimized within the range of 37–42 °C, and the reaction time was tested at 5 min intervals from 5 to 25 min to determine the optimal temperature and minimum time required to obtain a clear positive signal. Finally, the sensitivity of the assay was evaluated under the established optimal conditions by determining the limit of detection (LoD). A stock solution of *A. japonica* gDNA was serially diluted 10-fold to prepare the template DNA concentrations ranging from 10 to 0.1 ng, and the lowest amount of DNA that could be reliably detected was determined as the LoD through three replicate experiments.

## 3. Results

### 3.1. Primer Specificity and Accuracy

The MIRA assay achieved a high specificity, attributable to its underlying pair of primers, which were designed to bind selectively only to the mtDNA of *A. japonica*. Through multiple sequence alignment analysis, a target region with distinct sequence differences from other *Anguilla* species was selected. Specifically, the binding site for the forward primer (AJ-MIRAF) had a minimum of two and a maximum of six nucleotide variations compared to the other eel species (Figure 1). The reverse primer (AJ-MIRAR) binding site had a minimum of four and maximum of seven nucleotide variations (Figure 2). Crucially, the 3′ ends of the primers, which are the most critical for determining amplification specificity, were designed to be mismatched with the sequences of all non-target species included in this study, thereby fundamentally preventing non-specific amplification.

### 3.2. Specificity of the A. japonica MIRA-LFA Assay

To evaluate the species specificity of the MIRA-LFA, experiments were conducted using DNA from five major eel species, including *A. japonica*. A positive result, indicated by the clear appearance of both C and T lines, was observed only for the *A. japonica* sample (Figure 3). For the four non-target eel species (*A. anguilla*, *A. rostrata*, *A. bicolor pacifica*, and *A. marmorata*), only the C line appeared on all LFA strips, confirming that both the experimental procedure and the strips had functioned correctly. This result demonstrates that the MIRA-LFA developed in this study can distinguish *A. japonica* DNA from that of the other tested *Anguilla* species with extremely high specificity and without cross-reactivity.

### 3.3. Optimization of Amplification Temperature and Time

The performance of the assay was evaluated at different temperatures and reaction times to optimize its speed and field applicability. A very rapid reaction was observed at 39 °C, the optimal active temperature for the MIRA enzyme system. A distinct positive signal (T line) formed on the LFA strip after only 5 min of reaction time, and a clearer T line was obtained after 10 min (Figure 4a). The reaction was significantly slower at ambient temperature in an unheated environment (24 °C), with no detectable signal appearing within 10 min. A faint T line began to appear after 15 min, and a sufficiently clear signal for definitive visual identification formed after 20 min (Figure 4b). These results confirm that rapid identification is possible within just 5 min using a portable block heater, and that the test can be completed within 20 min at ambient temperature without a heating device, demonstrating its high field applicability.

### 3.4. Analytical Sensitivity

The LoD of the MIRA-LFA was evaluated under optimal temperature and reaction-time conditions. The assay demonstrated very high sensitivity when conducted at 39 °C for 10 min, reliably detecting as little as 0.1 ng/μL of *A. japonica* gDNA. This result indicates that reliable identification is possible even from samples containing minute amounts of DNA. For field situations requiring faster analysis, the LoD was determined to be 10 ng/μL after a 5 min reaction. While a positive signal from 10 ng/μL of gDNA at ambient temperature (24 °C) was first detectable at 15 min, a 20 min incubation was required to obtain an unambiguous result suitable for reliable identification. These results demonstrate that users can adopt a flexible analytical strategy, ensuring either the highest sensitivity (0.1 ng/μL at 39 °C for 10 min) or highest speed (10 ng/μL at 39 °C for 5 min), depending on the field situation (Table 3).

## 4. Discussion

The conservation of endangered species and the enforcement of international trade regulations such as CITES are critically dependent on the availability of accurate species identification technologies [27]. The MIRA-LFA system developed in this study addresses this challenge by providing a rapid, highly accurate identification method for *A. japonica* that effectively overcomes the field application limitations of conventional laboratory-based assays.

The primary advantages of the proposed MIRA-LFA are its speed and convenience. A visual result can be obtained from a DNA sample within 10 min at the optimal temperature of 39 °C, representing a significant time reduction compared to PCR-based methods that take several hours. Furthermore, the choice of MIRA over the more common RPA platform offers distinct advantages for field diagnostics. The technology’s use of proprietary enzymes is designed to enhance reagent stability, making the assay more robust to variable transport and storage conditions. Concurrently, its specific probe system minimizes non-specific amplification, a known challenge in some isothermal methods, thereby increasing the trustworthiness of the results obtained outside of a controlled laboratory environment [28]. This speed is competitive compared to other isothermal amplification techniques, such as loop-mediated isothermal amplification [29]. Furthermore, the reaction can be performed with a simple portable heating device or even body heat, eliminating the need for an expensive thermal cycler, and the intuitive result confirmation via an LFA strip eliminates the need for separate analytical equipment or complex data interpretation processes. This approach reduces costs and greatly enhances the accessibility of the diagnostic test, allowing non-experts to use it easily in the field. These features make the MIRA-LFA an ideal model that meets the ASSURED (affordable, sensitive, specific, user-friendly, rapid and robust, equipment-free, and deliverable) criteria for on-site diagnostics [30].

This technological advancement has the potential to bring about a practical change in the enforcement of eel trade regulations. Current enforcement relies on a reactive model where suspicious samples are sent to a laboratory for analysis, a process that can take days. This significant time lag means that by the time a species is identified, the illegal products may have already entered the complex distribution network, rendering enforcement ineffective [31]. However, the MIRA-LFA system overcomes this fundamental limitation through its core advantages of speed and on-site applicability. By providing a clear visual result in under 10 min without the need for a laboratory, it closes the critical time gap between suspicion and confirmation. This shift empowers frontline personnel such as customs inspectors, fisheries officers, and market regulators to move from a passive, forensic approach to proactive, on-the-spot enforcement. They can instantly verify the species of a shipment, enabling the immediate seizure of illegal cargo or the swift clearance of legitimate products, thus preventing illicit goods from ever entering the market [32]. Consequently, the widespread deployment of this rapid and accessible tool fundamentally alters the risk landscape for illegal operators. The increased probability of immediate detection at any point in the supply chain acts as a powerful deterrent, disrupting established trafficking routs and undermining the business models that rely on slow, lab-based verification.

The results of this study are consistent with recent research trends demonstrating the expanding application of RPA-LFA technology in the field of food authenticity. Similar assays have been successfully developed for species identification in various food products such as octopus and red meat [33,34], implying that the MIRA-LFA platform is a powerful and versatile tool for combating food fraud.

However, this study has several limitations, which imply directions for future research. First, the developed assay is a single-plex system targeting only *A. japonica*. The development of a multiplex MIRA-LFA system capable of simultaneously detecting CITES-regulated European eels or other major substitute species on a single strip would represent a more powerful on-site enforcement tool. Second, this study used a commercial kit for DNA extraction, which is a standard laboratory procedure. To create a true sample-to-answer on-site kit, it is essential to integrate a rapid DNA extraction method, such as thermal lysis or a simple buffer-based approach, with the MIRA-LFA. Third, the current two-step process requires opening the reaction tube, creating a significant risk of carryover contamination that can lead to false-positive results in a field setting. Future research should focus on developing a sealed, all-in-one cartridge system that integrates both amplification and detection steps to eliminate this risk. Finally, the robustness and practicality of the assay must be verified conclusively through large-scale validation studies conducted by non-expert users in actual field environments, such as ports, airports, and seafood markets.

## 5. Conclusions

This study developed an MIRA-LFA system as a powerful tool to address challenges in eel conservation and trade regulation. The assay combines high specificity (100%) and sensitivity (0.1 ng/μL) with a rapid detection time of under 10 min, in a user-friendly, equipment-free format. This technology bridges the gap between laboratory accuracy and field applicability, and provides an effective on-site solution for regulatory bodies and stakeholders to combat seafood fraud. The novel MIRA-LFA platform represents a significant advancement toward achieving the transparent and sustainable management of vulnerable eel populations.

## Figures and Tables

**Figure 1 foods-14-03100-f001:**
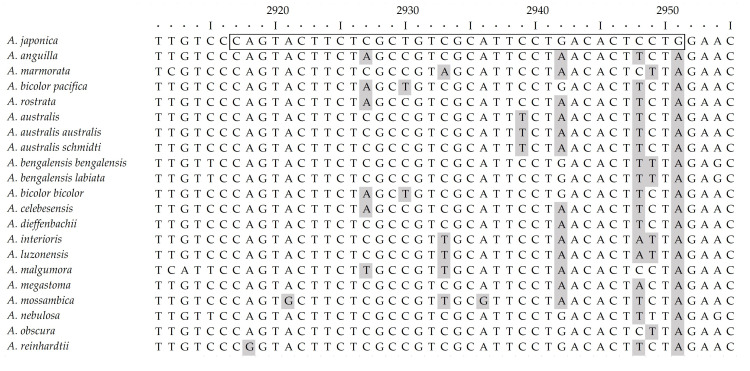
Alignment of the target sequence for the forward primer AJ-MIRAF to demonstrate its specificity for *A. japonica* via mismatches with other *Anguilla* species.

**Figure 2 foods-14-03100-f002:**
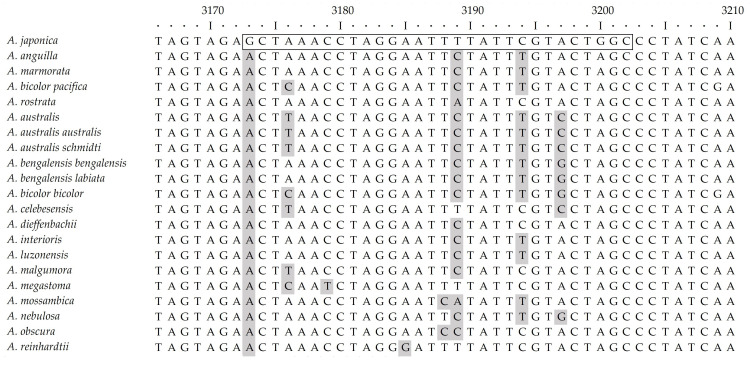
Alignment of the target sequence for the reverse primer AJ-MIRAR to demonstrate its specificity for *A. japonica* via mismatches with other *Anguilla* species.

**Figure 3 foods-14-03100-f003:**
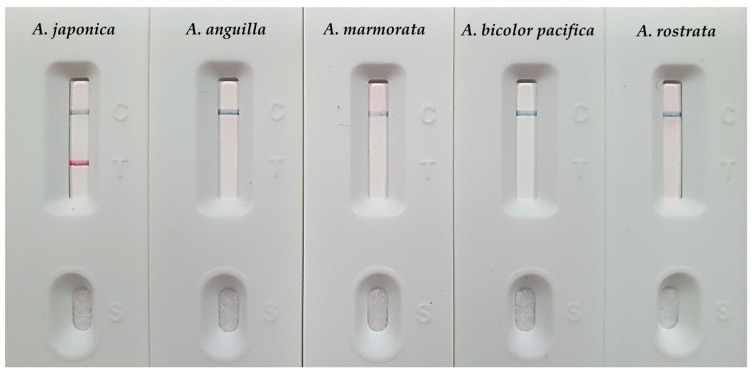
Evaluation of MIRA-LFA species specificity by testing its reactivity against genomic DNA from the target species, *A. japonica*, and four non-target *Anguilla* species.

**Figure 4 foods-14-03100-f004:**
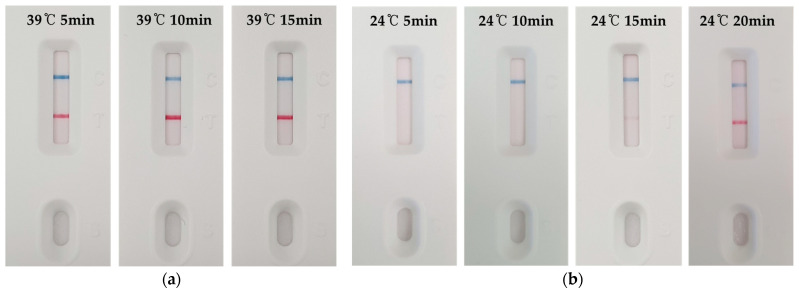
Determination of optimal reaction conditions for the MIRA-LFA by assessing its performance across a time course at (**a**) 39 °C and (**b**) ambient temperature (24 °C).

**Table 1 foods-14-03100-t001:** Primer sequences designed for the specific MIRA-LFA detection of *A. japonica*.

Primer Name	Sequence (5′ to 3′)	Modification	Length
AJ-MIRAF	CAGTACTTCTCGCTGTCGCATTCCTGACACTCCTG	5′-FAM	35 bp
AJ-MIRAR	GCCAGTACGAATAAAATTCCTAGGTTTAGC	5′-Biotin	30 bp

**Table 2 foods-14-03100-t002:** *Anguilla* species panel used to validate the specificity of the *A. japonica* MIRA-LFA.

Species	Accession Number	Number of Data
*Anguilla japonica*	NFIRD-FI-TS-0073097~0073104	8
*Anguilla anguilla*	NFRDI-FI-TS-0073157~0073164	8
*Anguilla rostrata*	NFRDI-FI-TS-0077939~0077940	2
*Anguilla bicolor pacifica*	NFRDI-FI-TS-0075207~0075214	8
*Anguilla marmorata*	NFRDI-FI-TS-0073247~0073254	8

**Table 3 foods-14-03100-t003:** Determination of the analytical sensitivity (limit of detection) of the MIRA-LFA using serially diluted *A. japonica* genomic DNA.

DNA Concentration	39 °C			24 °C		
	5 min	10 min	15 min	15 min	20 min	25 min
10 ng/μL	++	+++	+++	+	++	++
1 ng/μL	−	+++	+++	−	−	−
0.1 ng/μL	−	+++	+++	−	−	−

Signal intensity was graded as follows: −, no signal; +, weak signal; ++, moderate signal; +++, strong signal.

## Data Availability

The original contributions presented in the study are included in the article/Appendix A, further inquiries can be directed to the corresponding author.

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
