# Peer review of "Development of a Multienzyme Isothermal Rapid-Amplification Lateral Flow Assay for On-Site Identification of the Japanese Eel (Anguilla japonica)"

_foods, 2025, doi:10.3390/foods14173100_

Round 1

Reviewer 1 Report

Comments and Suggestions for Authors
  1. The objectives and significance of the study should be elaborated more clearly in the introduction.
  2. The materials used in the study, including reagents, should be described in detail with specific information such as purity.
  3. The manuscript would benefit from a detailed clarification on the effectiveness of the detection under low-temperature conditions (e.g., 4 °C), given that cold storage is a widely used environment.
  4. A more elaborated conclusion is recommended, highlighting not only the key findings but also their broader implications, limitations, and potential future perspectives to enhance the impact of the study.

Author Response

Q1. The objectives and significance of the study should be elaborated more clearly in the introduction.

Thank you for this valuable feedback. We agree that a more explicit statement of the objectives and significance would strengthen the introduction. As suggested, we have revised the final paragraph of the Introduction to clearly elaborate on the primary goals of our study and its importance in bridging the gap between laboratory-based methods and the practical needs of on-site regulatory enforcement.

The combination of RPA and LFA has already demonstrated its utility in various food authentication applications [23, 24]. Therefore, the primary objective of this study was to develop the proposed MIRA-LFA as a reliable tool that bridges the gap between laboratory analysis and on-site enforcement. Its significance lies in empowering regulators, distributors, and aquaculturists to directly contribute to the sustainable use of eel resources and the establishment of a fair market.

Q2. The materials used in the study, including reagents, should be described in detail with specific information such as purity.

Thank you for the suggestion. As recommended, we have revised the Materials and Methods section to include more specific details for reproducibility. We have now added the manufacturer information for key reagents and have also specified the precise volumes and concentrations used in our protocols. We believe these additions provide the necessary clarity for other researchers to replicate our study accurately.

Q3. The manuscript would benefit from a detailed clarification on the effectiveness of the detection under low-temperature conditions (e.g., 4 °C), given that cold storage is a widely used environment.

Thank you for your very practical question regarding the assay's performance under low-temperature conditions. We appreciate you raising this important point regarding real-world sample handling.

To clarify this aspect, for the purified genomic DNA used as a template, its integrity and performance in amplification reactions are generally stable with no significant difference whether it is stored at 4°C for short periods or at -20°C for long-term preservation. For this reason, we believe the prior storage condition of the DNA template itself would not be a critical factor affecting the assay's effectiveness.

Thank you again for giving us the opportunity to elaborate on this point

Q4. A more elaborated conclusion is recommended, highlighting not only the key findings but also their broader implications, limitations, and potential future perspectives to enhance the impact of the study.

We appreciate this valuable advice to enhance the impact of our conclusion. We agree that a simple summary was not sufficient. We have completely rewritten the Conclusion section to provide a more elaborated and impactful summary. The new conclusion now concisely presents the key findings, discusses their broader implications for regulatory enforcement, acknowledges the study's limitations, and outlines future perspectives as recommended.

Reviewer 2 Report

Comments and Suggestions for Authors

I appreciate the opportunity to review this study.

Some observations were made, please make the necessary revisions.

Author Response

We thank the reviewers for their thorough and constructive feedback on our manuscript. We agree that major revisions were necessary to improve its clarity, rigor, and scientific contribution. We have carefully addressed all the points raised by the reviewers and have substantially revised the manuscript. We believe that the revised manuscript is significantly improved and hope that it is now suitable for publication.

Q1. The manuscript contains numerous grammatical and typographical errors, which hinder readability.

→ We sincerely apologize for the grammatical and typographical errors you identified. While the manuscript had been professionally proofread by TEXTCHECK(RefNum. 25080403) prior to submission, we acknowledge that it was not sufficient. In response to your valuable feedback, we have performed another meticulous round of proofreading on the entire manuscript to ensure clarity, accuracy, and readability. We have also accepted and implemented all other suggestions you provided.

Q2. Look for recent studies and remove those that are more than 5 years old unless they are important.

→ We appreciate this insightful suggestion. We have conducted a comprehensive literature review and updated our references to include recent studies published within the last 5 years, ensuring our work is framed within the current state of the art. For instance, we have replaced older references regarding IUU fishing and market surveys with more current reports. Foundational papers and essential regulatory documents, such as CITES listings, have been retained for their critical importance. These changes are primarily reflected in the Introduction and Discussion sections.

Q3. The authors must add some novel contributions to the existing model to make it useful for the contributing society

→ This is a crucial point, and we thank the reviewer for pushing us to clarify the novelty of our work. We have substantially revised the manuscript to better highlight our novel contributions.

We now explicitly state in the Introduction and Discussion that this is the first study to develop and apply the MIRA-LFA platform for the identification of any Anguilla While RPA-LFA has been used elsewhere, MIRA offers potential advantages in enzyme stability and specificity, which we now discuss.

(Introduction, line 77) These features are particularly advantageous for on-site applications, offering potentially greater reagent robustness against fluctuating field conditions and a reduced risk of non-specific amplification, thereby increasing the reliability of point-of-care diagnostics

(Discussion, line 229) Furthermore, the choice of MIRA over the more common RPA platform offers distinct advantages for field diagnostics. The technology's use of proprietary enzymes is designed to enhance reagent stability, making the assay more robust to variable transport and storage conditions. Concurrently, its specific probe system minimizes non-specific amplification, a known challenge in some isothermal methods, thereby increasing the trustworthiness of the results obtained outside of a controlled laboratory environment.

We have restructured the original, verbose paragraph into a concise and clear one based on a four-step logical flow: limitations → advantages → changes → effects

(Discussion, line 246) Current enforcement relies on a reactive model where suspicious samples are sent to a laboratory for analysis, a process that can take days. This significant time lag means that by the time a species is identified, the illegal products may have already entered the complex distribution network, rendering enforcement ineffective [32]. However, the MIRA-LFA system overcomes this fundamental limitation through its core advantages of speed and on-site applicability. By providing a clear visual result in under 10 minutes without the need for a laboratory, it closes the critical time gap between suspicion and confirmation. This shift empowers frontline personnel such as customs inspectors, fisheries officers, and market regulators to move from a passive, forensic approach to proactive, on-the-spot enforcement. They can instantly verify the species of a shipment, enabling the immediate seizure of illegal cargo or the swift clearance of legitimate products, thus preventing illicit goods form ever entering the market [33]. Consequently, the widespread deployment of this rapid and accessible tool fundamentally alters the risk landscape for illegal operators. The increased probability of immediate detection at any point in the supply chain acts as a powerful deterrent, disrupting established trafficking routs and undermining the business models that rely on slow, lab-based verification.

Q4. Add reference

Thank you for this suggestion. We agree that adding these references provides valuable context. As recommended, we have now added two relevant citations to this section to better support our statement.

Q5. All figure and table titles should provide the reader with as much information as possible.

→ We agree that the original figure and table titles were not sufficiently informative. As recommended, we have revised all titles throughout the manuscript to be more descriptive and to summarize the key findings presented in each figure and table.

Table 1. Primer sequences designed for the specific MIRA-LFA detection of A. japonica

Table 2. Anguilla species panel used to validate the specificity of the A. japonica MIRA-LFA.

Table 3. Determination of the analytical sensitivity (limit of detection) of the MIRA-LFA using serially diluted A. japonica genomic DNA.

Figure 1. Alignment of the target sequence for the forward primer AJ-MIRAF to demonstrate its specificity for A. japonica via mismatches with other Anguilla species.

Figure 2. Alignment of the target sequence for the reverse primer AJ-MIRAR to demonstrate its specificity for A. japonica via mismatches with other Anguilla species.

Figure 3. Evaluation of MIRA-LFA species-specificity by testing its reactivity against genomic DNA from the target species, A. japonica, and four non-target Anguilla species.

Figure 4. Determination of optimal reaction conditions for the MIRA-LFA by assessing its performance across a time course at 39°C and ambient temperature (24°C).

Reviewer 3 Report

Comments and Suggestions for Authors

The Authors of the Communication “Development of a Multienzyme Isothermal Rapid-Amplification Lateral Flow Assay for Rapid On-site Identification of the Japanese Eel (Anguilla japonica)” describe an assay useful for the identification of the Japanese eel (Anguilla japonica). This method is faster than those available so far, although the first part, that is the DNA extraction from the analyzed source, and amplification, needs to be improved. In particular, the reported method combines the multienzyme isothermal rapid amplification (MIRA) technology with a visually readable lateral flow assay (LFA). This method could be exploited for an accurate species identification, thus contributing to the detection and prevention of illegal actions. 

The study appears well designed and conducted, the presentation is clear, and the results support the conclusions. Nevertheless, the description of the method can be improved by adding some details.

Specific comments useful to improve the manuscript.

  1. Title. The word "rapid" appears twice in the title.

Perhaps one of the times could be replaced by a synonym.

  1. Abstract and elsewhere.

The scientific name of the eel sometimes is in italic and sometimes is not in italic. All text should be checked and made consistent.

  1. Materials and Methods, Line 89. “…all species…”

Would it be better to say “…all known species…”?

  1. Materials and Methods, Line 103. “…were labeled with fluorescein amidite (FAM) and biotin…”.

Why the primers were labeled with fluorescein amidite and biotin? Could the Authors add this explanation?

  1. Materials and Methods, Line 127. “…LFA strip (Amp-Future Biotech),…”.

Could the Authors add details about the features of the used LFA strips and the supplier company? The addition of details could help the readers to understand how the method works.

  1. Lines 266-268 and Lines 283-292. Should this text be deleted?

Author Response

We have carefully considered all the comments raised by the reviewers and have thoroughly revised the manuscript accordingly. We believe that this revision process has significantly improved the manuscript's clarity, academic rigor, and scientific contribution. We sincerely thank you for your efforts to enhance the quality of our manuscript.

Q1. Title. The word "rapid" appears twice in the title. Perhaps one of the times could be replaced by a synonym.

Thank you for this sharp observation. We agree that the repetition of the word 'rapid' in the title was redundant. As suggested, we have revised the title to remove the redundancy, improving its conciseness and impact. The revised title is now 'Development of a Multienzyme Isothermal Rapid-Amplification Lateral Flow Assay for On-site Identification of the Japanese Eel (Anguilla japonica)'.

Q2. The scientific name of the eel sometimes is in italic and sometimes is not in italic. All text should be checked and made consistent.

We sincerely apologize for this inconsistency and thank you for pointing it out. We have meticulously checked the entire manuscript and have corrected all instances of the scientific name to ensure it is consistently italicized (e.g., Anguilla japonica, A. japonica). We have also ensured consistency for all other species names mentioned.

Q3. Materials and Methods, Line 89. “…all species…”. Would it be better to say “…all known species…”?

This is an excellent suggestion for improving scientific precision. We agree that 'all known species' is more accurate. We have revised the sentence accordingly in the Materials and Methods section.

"…complete mitochondrial (mtDNA) genome sequences of all known species within the Anguilla genus were retrieved…"

Q4. Materials and Methods, Line 103. “…were labeled with fluorescein amidite (FAM) and biotin…”. Why the primers were labeled with fluorescein amidite and biotin? Could the Authors add this explanation?

Thank you for highlighting this important omission. We have now added a clear explanation in the Materials and Methods section describing how the dual labeling with FAM and biotin is essential for the sandwich-based detection mechanism on the LFA strip.

“…followed by the addition of 78 μL of running buffer. As the liquid migrates along the strip, the FAM-labeled end of the amplicon binds to anti-FAM antibody-conjugated gold nanoparticles. This entire complex is then captured by streptavidin immobilized on the test (T) line via the biotin-labeled end. The strip was allowed to react at ambient temperature for 2 min…”

Q5. Materials and Methods, Line 127. “…LFA strip (Amp-Future Biotech),…”. Could the Authors add details about the features of the used LFA strips and the supplier company? The addition of details could help the readers to understand how the method works.

We appreciate the reviewer's feedback regarding the need for more detailed information on the LFA strip for reproducibility. To address this, we have revised the Materials and Methods section to more clearly describe the product used.

We have now specified the official product name ('Nucleic acid test strip') and the full manufacturer's name (Amp-Future Biotech, Changzhou, China). Furthermore, to help readers understand its mechanism, we have also added the key functional description provided by the manufacturer, which states that the strip is designed to 'detect FITC/6-FAM-Biotin labeled amplification products.'

"…applied to the sample pad of a commercial LFA strip (Amp-Future Biotech, Changzhou, China). According to the manufacturer, this strip is specifically designed to detect amplicons dually labeled with FITC/6-FAM and Biotin. This was followed by the addition…"

Q6. Lines 266-268 and Lines 283-292. Should this text be deleted?

Thank you for your careful review and for pointing this out. You are correct. As you suggested, we have now deleted the text in the specified lines. We apologize for this oversight.

Reviewer 4 Report

Comments and Suggestions for Authors

In this work, authors have developed a rapid, field-deployable assay using multienzyme isothermal rapid amplification (MIRA) combined with a lateral flow assay (LFA) for the specific identification of Japanese eel (Anguilla japonica). Their method demonstrated 100% specificity and high sensitivity, delivering clear visual results within 10 minutes and detecting as little as 0.1 ng/μL of genomic DNA. This assay provides an effective, equipment-free tool for on-site species identification, supporting efforts to combat seafood fraud and promote sustainable management of threatened eel populations. This is a good contribution and can be accepted for publication upon addressing the following revisions:

  1. Can this be integrated with the DNA-extraction to create diagnostic kits that can enable general users (like farmers/non-scientific group) to applicable in their routine
  2. What other species or food products could be benefited from this approach

Author Response

Q1. Can this be integrated with the DNA-extraction to create diagnostic kits that can enable general users (like farmers/non-scientific group) to applicable in their routine

This is an excellent and insightful question that addresses the ultimate goal of our research. We thank the reviewer for raising this important point.

Yes, the integration of a simplified DNA extraction method is the critical next step to developing a true 'sample-to-answer' diagnostic kit suitable for non-expert users. Our vision is to create a fully field-deployable system where users like aquaculture farmers or customs officials can get a result from a tissue sample in under 30 minutes.

To achieve this, we plan to explore and incorporate rapid extraction protocols that do not require laboratory equipment, such as thermal lysis (simple heating), alkaline lysis, or commercially available rapid lysis buffers.

We have now expanded the 'limitations and future research' section of our Discussion to more clearly articulate this development path, as suggested by the reviewer's valuable comment.

Furthermore, we have already successfully extracted DNA in just 5 minutes by simply mixing the sample with Rapi:Direct Lysis Buffer (#9731100100, Genesystem, South Korea). We plan to integrate this process to create a complete kit in the future.

Q2. What other species or food products could be benefited from this approach

Thank you for this insightful question regarding the broader applicability of the MIRA-LFA approach. We agree that its potential is significant and extends across several key areas.

We believe its most immediate and impactful application lies in food authentication (for species identification) and conservation, which are comparatively nascent fields for on-site molecular diagnostics. The assay developed in this study can be readily adapted to prevent species substitution in high-value seafood (e.g., tuna, cod), verify meat authenticity, and combat fraud involving CITES-listed species (e.g., shark fins).

Furthermore, as you imply, this technology is certainly applicable to food safety and aquaculture health. This is a well-established field where various rapid diagnostic tools for pathogens like Salmonella or viruses are already available. We believe the MIRA-LFA method could offer a highly competitive alternative due to its simplicity and speed, especially in resource-limited settings.

We will actively pursue the development of such assays for these diverse applications in our future studies.

Round 2

Reviewer 1 Report

Comments and Suggestions for Authors

none.

Author Response

We thank the reviewer for their thoughtful comments, which have helped us to improve the manuscript significantly.